# Association between Serum Phosphate Levels and the Development of Aortic Stenosis in Patients Undergoing Hemodialysis

**DOI:** 10.3390/jcm10194385

**Published:** 2021-09-25

**Authors:** Miki Torigoe, Mineaki Kitamura, Kosei Yamaguchi, Takumi Uchino, Kenta Torigoe, Takashi Harada, Satoshi Funakoshi, Kazuko Yamamoto, Koji Maemura, Kiyoyuki Eishi, Hiroshi Mukae, Tomoya Nishino

**Affiliations:** 1Department of Nephrology, Nagasaki University Graduate School of Biomedical Sciences, Nagasaki 852-8523, Japan; sawamiki.114@gmail.com (M.T.); tugumasayamaguchi@gmail.com (K.Y.); ktorigoe@nagasaki-u.ac.jp (K.T.); tnishino@nagasaki-u.ac.jp (T.N.); 2Nagasaki Renal Center, Nagasaki 852-8102, Japan; godebisu1@icloud.com (T.U.); renaltharada@nagajin.jp (T.H.); satoshi2754@yahoo.co.jp (S.F.); 3Department of Respiratory Medicine, Nagasaki University Graduate School of Biomedical Sciences, Nagasaki 852-8523, Japan; kazukomd@nagasaki-u.ac.jp (K.Y.); hmukae@nagasaki-u.ac.jp (H.M.); 4Department of Cardiovascular Medicine, Nagasaki University Graduate School of Biomedical Sciences, Nagasaki 852-8523, Japan; maemura@nagasaki-u.ac.jp; 5Department of Cardiovascular Surgery, Nagasaki University Graduate School of Biomedical Sciences, Nagasaki 852-8523, Japan; keishi@nagasaki-u.ac.jp

**Keywords:** aortic stenosis, chronic hemodialysis, phosphatemia, prognosis, age

## Abstract

We aimed to investigate the factors associated with the development of aortic stenosis (AS) in patients undergoing hemodialysis (HD), and to elucidate the prognosis of HD patients with AS. Patients on HD that had also undergone echocardiography at Nagasaki Renal Center between July 2011 and June 2012 were included. Patients with AS at the time of inclusion were excluded. The diagnosis of AS was based on an annual routine or additional echocardiography. The patients were followed up until June 2021. The association between patient background and AS was also evaluated. Of the 302 patients (mean age, 67.4 ± 13.3 years; male, 58%; median dialysis history, 4.7 years), 60 developed AS and 10 underwent aortic valve replacement. A Cox proportional hazards model revealed that age (hazard ratio (HR), 1.07; 95% confidential interval (CI), 1.04–1.10; *p* < 0.001) and serum phosphate levels (HR, 1.40; 95%CI, 1.16–1.67, *p* < 0.001) were independent risk factors for developing AS. Incidentally, there was no significant mortality difference between patients with AS and those without (*p* = 0.53). Serum phosphate levels are a risk factor for developing AS and should be controlled. Annual echocardiography may contribute to the early detection of AS and improves the prognosis of patients undergoing HD.

## 1. Introduction

Patients on hemodialysis (HD) have various comorbidities, with aortic stenosis (AS) being one of the most critical complications. The high prevalence of severe AS in HD patients has been clearly characterized [1]. When AS becomes severe, symptoms of heart failure, such as breathlessness and palpitation, appear, and fainting occurs. Patients with symptomatic severe AS have a poor prognosis and require urgent treatment, such as aortic valve replacement [2]. Since AS is associated with aging [3,4], its incidence in dialysis patients is believed to increase with age [5]. The incidence rate of AS in dialysis patients is 3.3%/year [6], and the aortic valve area stenosis rate and progression are faster than those in the general population [7,8,9]. 

Chronic kidney disease–mineral bone disorders (CKD–MBD) are associated with cardiovascular diseases and arterial sclerosis in patients on HD [10,11]. For example, serum phosphate levels are associated with stroke [12] and peripheral artery disease [13] in patients undergoing HD. Although aging [14,15], sex [16], duration of HD [14], serum phosphate levels [14], and diabetes [16] are known to be associated with valvular calcification, little is known about the association between CKD–MBD parameters and the development of AS in HD patients. Similarly, high phosphate concentration is a risk factor for vascular calcification and cardiovascular diseases in dialysis patients. However, there is a paucity of studies that have investigated this relationship with the development of AS. We aimed to investigate the factors associated with the development of AS in patients undergoing HD, particularly CKD–MBD parameters. In addition, we set out to elucidate the prognosis of HD patients with AS, including those with mild AS. 

## 2. Materials and Methods

### 2.1. Patients

Patients who underwent maintenance HD at Nagasaki Renal Center between July 2011 and June 2012 were included in this study, as described previously [17]. Since the patients’ characteristics were collected in their birth months, including echocardiographic findings, patients who died or were transferred to other facilities before their birth months and those who initiated HD therapy after their birth months were excluded. Patients who did not undergo echocardiography during the inclusion period and those with pre-existing AS were also excluded.

### 2.2. Data Collection

Patient characteristics, such as age, sex, dialysis history, dialysis time, complications, blood pressure, and blood examination results, were collected during their birth months in the inclusion period. Data for serum phosphate levels were collected one year after the inclusion period. Echocardiography was routinely conducted in the birth months and was also performed when patients exhibited symptoms. 

Patients were diagnosed with AS mainly based on a maximum trans-aortic velocity >2.0 m/s [18] or aortic valve area <1.5 cm^2^. The severity of AS was defined according to the maximum trans-aortic velocity (>4.0 m/s: severe, 3.0–4.0 m/s: moderate, and 2.0–3.0 m/s: mild) [18]. Patients were followed up until 30 June 2021.

### 2.3. Statistical Analysis 

Categorical variables are presented as numbers (%), and continuous variables are presented as mean ± standard deviation. Non-normally distributed data are displayed as the median of the interquartile range. Continuous variables were compared using the *t*-test and Mann–Whitney U test, and categorical variables were evaluated using the chi-square test. Darbepoetin and epoetin beta pegol were transformed into epoetin alfa using a 1:200 ratio, as described previously [17]. Survival was evaluated using Kaplan–Meier analysis and log-rank test. The risk of developing AS was evaluated using multivariable Cox proportional regression models. Age, sex, dialysis history, serum corrected calcium, serum phosphate levels, and intact parathyroid hormone (iPTH) levels were included in the multivariable model (Model 1). Model 2 included parameters whose *p*-values were <0.1 in the univariate Cox regression model. To evaluate the correlation between serum phosphate levels during the inclusion period and those recorded one year after the inclusion period, linear regression model was used.

## 3. Results

Of the 441 patients undergoing HD during the inclusion period, patients whose annual routine examination data could not be collected during this period were excluded. Echocardiography was not conducted during the birth months of 12 patients during the inclusion period. Five patients underwent aortic valve replacement, and 20 were diagnosed with AS during the inclusion period. Therefore, 302 patients were included in the study. The patient flow is shown in Figure 1. The median observation period was 4.3 (1.9–9.2) years. During the observation period, 60 patients (19.9%) developed AS, and the cumulative incidence of AS is shown in Figure 2. Patients’ characteristics are shown in Table 1. The severity of AS was mild in 50, moderate in 9, and severe in 2 patients at first diagnosis. However, the AS progressed in 17 patients during the observation period, reaching a maximum severity of mild in 36 patients, moderate in 15, and severe in 10. Among them, 10 patients (seven with severe and three with moderate AS) underwent aortic valve replacement therapy. The median period between the first diagnosis of AS and surgery was 431 days (range, 209–1343 days). 

The Kaplan–Meier curves showed no significant difference between the patients who developed AS and those who did not (Figure 3a). Although there were no sudden deaths among the patients with severe AS, 2 of 36 patients with mild AS and 4 of 15 with moderate AS died suddenly during the observation period. Despite the severity of AS, the mortality rate of the patients who underwent aortic valve replacement was superior to that of those who did not (*p* < 0.001) (Figure 3b). (*p* < 0.001). Even twenty cases with pre-existing AS at the inclusion period were included (three cases received aortic valve replacement after the inclusion criteria); this tendency was also observed in survival analysis (*p* < 0.001). 

Univariate and multivariable Cox proportional models were used to assess the risk of AS. As shown in Table 2, age (Model 1: hazard ratio (HR), 1.07; 95% confidential interval (CI), 1.04–1.10; *p* < 0.001; and Model 2: HR, 1.06; 95% CI, 1.03–1.09; *p* < 0.001) and serum phosphate levels (Model 1: HR, 1.40; 95% CI, 1.16–1.67; *p* < 0.001, Model 2: HR, 1.44; 95% CI,1.21–1.70; *p* < 0.001) were independent risk factors for developing AS. Incidentally, the serum phosphate levels of 258 patients were available one year after the inclusion period, and there was a moderate correlation between serum phosphate levels during the inclusion period and those recorded one year after inclusion (r = 0.47, *p* < 0.001).

## 4. Discussion

We conducted a retrospective cohort study to elucidate the association between patient background and the development of AS. During the observation period, approximately 20% of patients developed AS, irrespective of severity. According to the Cox regression analyses, serum phosphate levels were associated with the development of AS. However, the prognosis of patients who developed AS was not poorer than those who did not develop AS.

The prevalence of AS among patients on HD is approximately 15–20%, with significant symptomatic AS found in 3–9% [14]. In this study, 20 patients were excluded because of pre-existing AS at the inclusion period, which accounted for almost 15% of the total cohort. The incidence of AS in patients on HD is higher than that in the general population, and some patients progress to severe AS that is accompanied by symptoms of angina, syncope, and heart failure [6,7,8,9]. The median time from the first diagnosis of AS to valve replacement was 431 days in this study, which was shorter than that in the general population [19].

Although CKD–MBD parameters are associated with valvular calcification and correlate with the severity of AS [20], little is known about their association with the development of AS in HD patients. CKD–MBD play a major role in arterial and valvular calcification in dialysis patients, but no association was found between AS onset and iPTH or serum calcium levels. Since valvular calcification reflects atherosclerosis and arterial calcification in HD patients [21], cardiovascular complications may be more prevalent in those with AS. In addition to valvular calcification, several factors, including age, duration of dialysis, phosphate and/or Ca×P product, oral calcium intake, use of vitamin D, diabetes, C-reactive protein, and low albumin, are generally associated with ectopic calcification [22]. The results of this study indicate that phosphate has a great effect on the development of AS. The management of serum phosphate should be prioritized as it is a modifiable factor that can be controlled. 

Several important factors need to be considered with regard to controlling the phosphate levels in the context of HD. Although a Japanese guideline recommended maintaining serum phosphate levels within 3.5–6.0 mg/dL [23], the optimal target range for preventing ectopic calcification would be narrower. A recent randomized control study elucidated the optimal phosphate level to prevent coronary artery calcifications [24], finding that strict control (3.5–4.5 mg/dL) achieved better outcomes than standard control (5.0–6.0 mg/dL). Although the optimal phosphate levels for preventing AS should be investigated in the future, strict phosphate control would be advantageous for patients on HD. Second, the type of phosphate binder should be considered. For example, the Kidney Disease Improving Global Outcomes guidelines recommended the use of calcium-free phosphate binders [10], which did not accelerate valvular calcification in previous studies [25,26,27]. In this study, there was no significant difference in the prescription rate of calcium carbonate between patients who did or did not develop AS; however, lanthanum carbonate was frequently prescribed in patients who developed AS despite their high serum phosphate levels. Finally, dietary guidance and ensuring adequate dialysis are alternative options for lowering serum phosphate levels. However, it should be noted that excessive dietary and phosphorus restrictions reduce protein intake and may worsen the prognosis [23]. Aging is another risk factor for developing AS, but excessive phosphate restrictions should be particularly avoided in elderly patients on HD. 

Although the precise mechanisms of AS development remain unknown, the trigger is believed to be damage to the valvular endothelium due to mechanical stress or other factors [11,28]. For instance, the accumulation of low-density lipoprotein (LDL) evokes valvular damage through oxidative LDL, which activates inflammatory cells. Valvular stromal cells are also activated and produce disordered collagen, causing fibrosis, valve thickening, and stiffening [29]. Valve calcification increases osteoblast activation, which, in turn, causes calcification in a vicious cycle [30]. Furthermore, high serum phosphate levels accelerate the transformation of vascular smooth muscle cells into osteoblasts via the fibroblast growth factor 23 and klotho axis [31].

Early detection and treatment before symptoms appear may improve the prognosis of patients on HD. Due to its non-invasiveness, annual Doppler echocardiography is recommended if the patients can tolerate aortic valve surgery [32]. Transcatheter aortic valve implantation (TAVI) is another option for HD patients, and its adoption, particularly in patients with various complications, is expected to increase. Therefore, an annual echocardiography will be recommended for almost all HD patients. Notably, the mortality of patients with AS did not differ from that of patients without AS in this study, suggesting that the annual echocardiographic survey may allow us to detect AS in the early stages, leading to favorable outcomes.

Ten patients who developed AS in this study underwent surgical aortic valve replacement, after which they all recovered successfully. Furthermore, they tended to achieve longer survival compared with patients across all disease severities who did not undergo aortic valve surgery. Nonetheless, patients on HD are at significant perioperative risk and exhibit higher postoperative mortality and infection rates [33]. TAVI is a new alternative to AS treatment in patients who are inoperable or have high surgical risk. Although studies have shown that the efficacy and safety of TAVI is equivalent to surgical aortic valve replacement in the general population [34,35], only a few studies have investigated TAVI in patients on HD [36,37,38]. The efficacy of TAVI in patients on HD will need to be elucidated in future large-scale studies. Incidentally, more than a quarter of patients diagnosed with moderate AS in this study died suddenly, possible due to progression to severe disease. Although early surgery leads to better outcomes than conservative observation in a general population with asymptomatic very severe AS [39], the optimal timing of follow-up echocardiography and valve replacement in patients on HD needs to be elucidated. 

This study has several limitations. First, blood examination results were only based on the inclusion period; thus, the serum phosphate levels would have likely changed during the observation period. In fact, there was only a moderate correlation between the serum phosphate levels during the inclusion period and those recorded one year after the inclusion period, suggesting that the serum phosphate levels fluctuated during the observation period. Second, echocardiography was routinely conducted in the birth months and when patients had symptoms. Therefore, this study could not determine the precise timing of AS development. Third, this study was conducted at a single center and almost all of the patients were Japanese. The results of this study may not be applicable to other patient populations. 

This study also had several strengths. In contrast to previous studies, we elucidated the risk of the development of AS based on our institution’s policy of routinely conducting annual echocardiographic examinations in patients undergoing HD. Additionally, we observed patients for a long period of time, which allowed us to elucidate the deterioration of AS over the course of time as well as the patients’ prognoses after surgery.

## 5. Conclusions

In conclusion, serum phosphate levels may play an important role in the development of AS in HD patients. Particular attention should be paid to serum phosphate levels and routine annual echocardiography in clinical practice, which are useful for the early detection of AS and may help to improve patient prognosis.

## Figures and Tables

**Figure 1 jcm-10-04385-f001:**
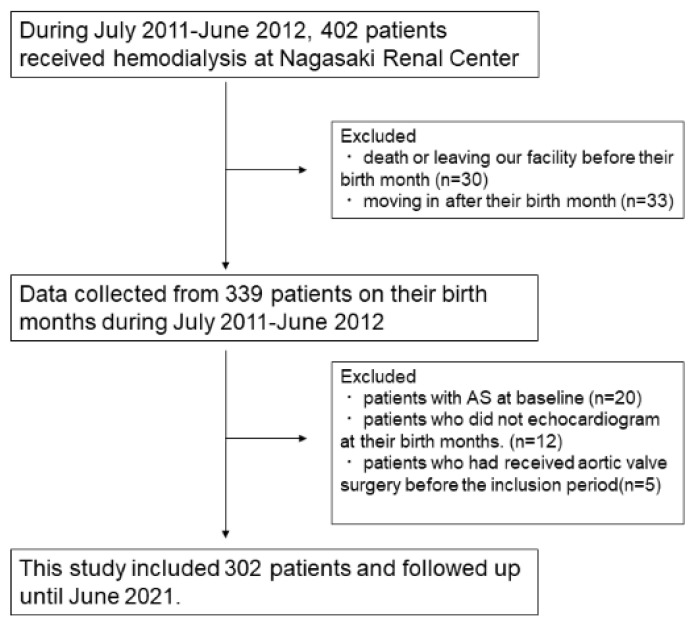
Patient flow.

**Figure 2 jcm-10-04385-f002:**
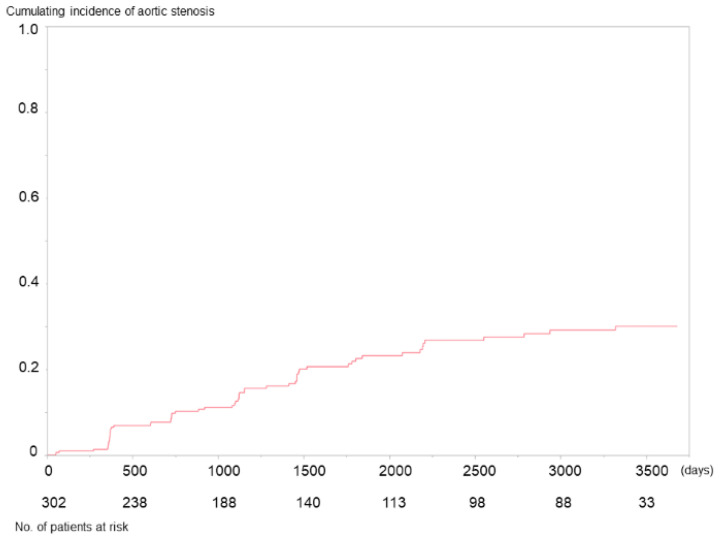
Cumulative incidence of aortic stenosis. The number below the graph shows the number of patients at risk.

**Figure 3 jcm-10-04385-f003:**
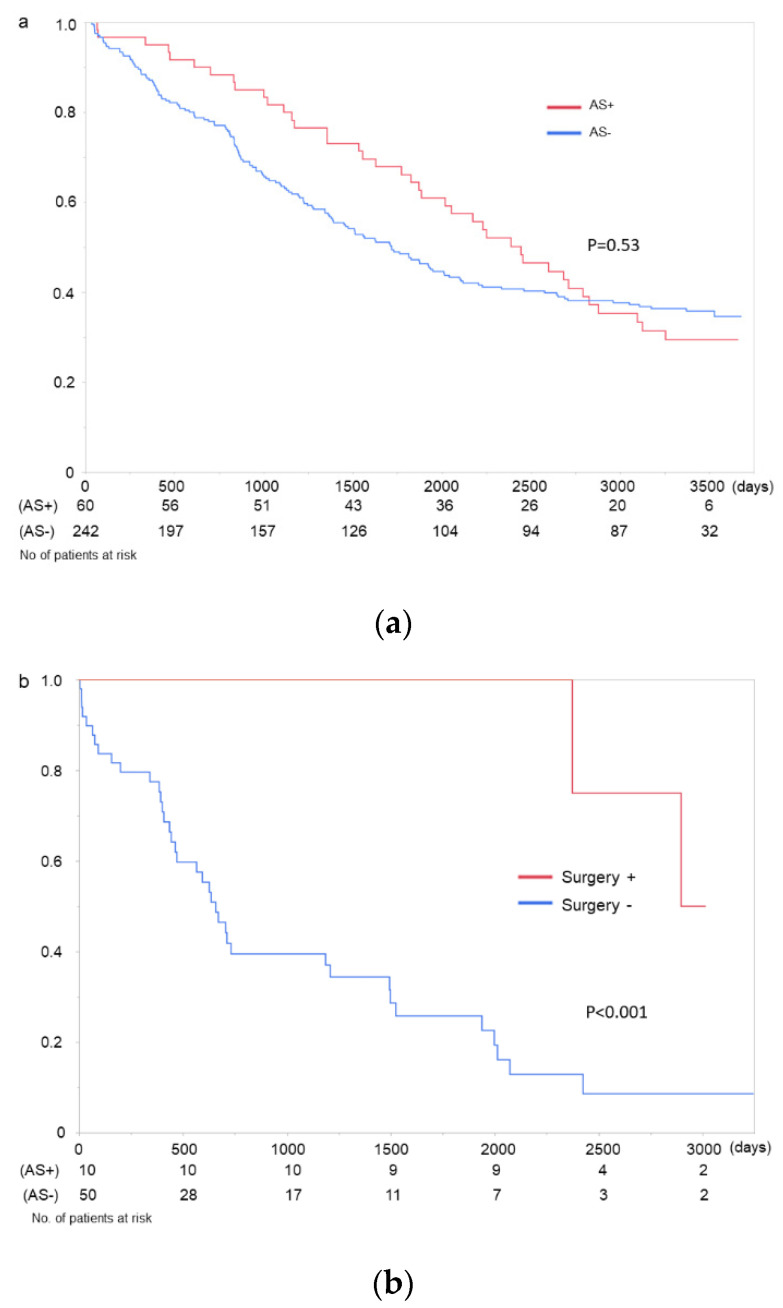
Kaplan–Meier survival analyses. (**a**) Patients who developed aortic stenosis during the follow-up period and those who did not. There were no significant differences between the two groups. The figures below the graph show the number of patients at risk of death according to the development of aortic stenosis. AS+: patients who developed aortic stenosis; AS−: patients who did not develop aortic stenosis. (**b**) Survival analysis from the first diagnosis of aortic stenosis. Patients who underwent aortic valve replacement showed a better outcome than those who did not, including those with mild aortic stenosis. The figures below the graph show the number of patients at risk of death according to undergoing surgery. S+: patients undergoing surgery, S−: patients not undergoing surgery.

**Table 1 jcm-10-04385-t001:** Patient characteristics according to the development of aortic sclerosis during the observation period.

	AS (n = 60)	Without AS (n = 242)	*p*-Value
Age (years)	69.6 ± 11.0	65.7 ± 13.4	0.04
Male (%)	55.0	58.7	0.61
Dialysis vintage ^a^ (years)	5.3 (1.9–9.0)	4.8 (2.0–10.5)	0.58
Dialysis time ^a^ (h)	4 (3–4)	4 (3–4)	0.64
Hypertension (%)	91.7	84.2	0.12
Diabetes mellitus (%)	33.3	35.1	0.79
Ischemic heart disease (%)	40.0	33.5	0.35
Cerebral hemorrhage (%)	3.3	7.4	0.22
Cerebral infarction (%)	23.3	24.4	0.87
Arteriosclerosis obliterans (%)	16.7	17.4	0.90
Cardiothoracic ratio (%)	52.6 ± 5.4	51.7 ± 5.7	0.26
Dry weight (kg)	51.7 ± 8.9	52.7 ± 11.5	0.54
Systolic blood pressure (mmHg)	154 ± 23	149 ± 25	0.18
Left ventricular ejection fraction (%)	66 ± 9	65 ± 11	0.37
Hemoglobin (g/dL)	11.1 ± 1.2	10.8 ± 1.3	0.13
Ferritin^a^ (ng/mL)	57.5 (24.0–197.9)	69.3 (25.7–178.9)	0.73
Transferrin saturation (%)	26.4 ± 14.4	25.1 ± 14.1	0.53
Albumin (g/dL)	3.6 ± 0.3	3.6 ± 0.4	0.31
Corrected calcium (mg/dL)	9.3 ± 0.7	9.2 ± 0.7	0.53
Phosphate (mg/dL)	6.1 ± 1.7	5.5 ± 1.6	0.02
Intact parathyroid hormone (pg/mL)	91 (24–154)	67 (27–157)	0.68
Alkaline phosphatase (IU/L)	252 (179–342)	250 (192–343)	0.55
Blood urea nitrogen (mg/dL)	71.9 ± 16.7	67.7 ± 17.4	0.10
Creatinine (mg/dL)	11.0 ± 2.7	10.3 ± 3.4	0.19
Total cholesterol (mg/dL)	163 ± 31	162 ± 37	0.83
Triglycerides (mg/dL)	82 (68–135)	92 (65–130)	0.70
C-reactive protein ^a^ (mg/dL)	0.11 (0.07–0.43)	0.18 (0.07–0.48)	0.39
ESA ^a^ (IU/week)	4250 (1625–8000)	4500 (2000–8000)	0.85
Calcium carbonate (%)	41.7	50.0	0.25
Lanthanum carbonate (%)	50.0	28.5	0.002
Sevelamer (%)	1.7	3.7	0.39
Cinacalcet (%)	20.0	17.0	0.59
Vitamin D (%)	70.0	66.1	0.56
Kt/V	1.32 ± 0.38	1.35 ± 0.40	0.61

Data are expressed as mean ±standard deviation or ^a^ median (interquartile range). Comparisons were performed using the *t*-test or Mann–Whitney U test. AS, aortic stenosis; ESA, erythropoiesis-stimulating agents.

**Table 2 jcm-10-04385-t002:** Univariable and multivariable Cox proportional regression models for the risk of developing aortic stenosis.

	Univariate			Model 1			Model 2		
	HR	95% CI	*p*-Value	HR	95% CI	*p*-Value	HR	95% CI	*p*-Value
Age/years	1.05	1.03–1.08	<0.001	1.07	1.04–1.10	<0.001	1.06	1.03–1.09	<0.001
Male vs. female	0.97	0.59–1.62	0.92	1.06	0.63–1.78	0.83			
HD vintage/year	0.98	0.94–1.01	0.15	0.99	0.96–1.03	0.91			
HD time/h	0.58	0.36–0.92	0.02				1.02	0.59–1.71	0.94
DM	1.18	0.69–2.03	0.54	0.97	0.53–1.76	0.91			
IHD history	1.39	0.83–2.32	0.22						
Stroke history	1.05	0.58–1.88	0.88						
CTR/%	1.07	1.02–1.12	0.004				1.04	0.99–1.10	0.08
DW/kg	0.98	0.95–1.00	0.07				0.99	0.96–1.02	0.56
SBP/10 mmHg	1.07	0.97–1.19	0.19						
Hb/g/dL	1.11	0.91–1.35	0.31						
Alb/g/dL	0.57	0.29–1.15	0.12						
cCa/g/dL	1.16	0.80–1.63	0.44	1.31	0.93–1.80	0.12			
P/mg/dL	1.16	0.98–1.35	0.08	1.40	1.16–1.67	<0.001	1.44	1.21–1.70	<0.001
iPTH/10 pg/mL	1.00	0.98–1.02	0.77	1.00	0.98–1.02	0.94			
BUN/10 mg/dL	1.07	0.91–1.26	0.40						
Cr/mg/dL	0.97	0.90–1.05	0.47						

Model 1: includes clinical important factors and chronic kidney disease–mineral bone disorders associated factors. Model 2: includes factors whose *p* values in the univariate model were < 0.10. HR, hazard ratio; HD, hemodialysis; DM, diabetes mellitus; IHD, ischemic heart disease; CTR, cardiothoracic ratio; DW, dry weight; SBP, systolic blood pressure; Hb, hemoglobin; Alb, albumin; cCa, corrected calcium; P, phosphate; iPTH, intact parathyroid hormone, BUN, blood urea nitrogen; Cr, creatinine

## Data Availability

The data underlying this article will be shared upon reasonable request from the corresponding author.

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
