# Peer review of "Association between Serum Phosphate Levels and the Development of Aortic Stenosis in Patients Undergoing Hemodialysis"

_jcm, 2021, doi:10.3390/jcm10194385_

Round 1
Reviewer 1 Report
Review.
The authors of the manuscript entitled Association between serum phosphate levels and the development of aortic stenosis in patients undergoing hemodialysis presented interesting topic.
The introduction to the topic is well written but it should be a bit extended. High phosphate concentration is a risk factor for vascular calcification and cardiovascular diseases in dialysis patients. There is little work that has investigated this relationship with aortic stenosis.
The Methodology and Results are clearly described.
The most important limitations were presented by the authors themselves. The lack of further measurements of phosphate concentration and other biochemical parameters is a big problem because it could change the statistics - therefore the results must be carefully interpreted.
In conclusion, it should rather be written: serum phosphate levels may play an important role in the development 234 of AS in HD patients.
Also, it is know that the concentration of phosphate in hemodialysis patients depends on the dialysis dose. There is no information on dialysis in the article, what was the treatment regimen?What is the adequacy of dialysis (Kt / V?).You need to add this information.
Author Response
The authors of the manuscript entitled Association between serum phosphate levels and the development of aortic stenosis in patients undergoing hemodialysis presented interesting topic.
The introduction to the topic is well written but it should be a bit extended.
High phosphate concentration is a risk factor for vascular calcification and cardiovascular diseases in dialysis patients. There is little work that has investigated this relationship with aortic stenosis.
[Reply]
Thank you for your constructive comment. We have included the expression that you suggested on Lines 48–50 of the manuscript. It definitely enhances the presentation of the aim of this study.
The Methodology and Results are clearly described.
The most important limitations were presented by the authors themselves. The lack of further measurements of phosphate concentration and other biochemical parameters is a big problem because it could change the statistics - therefore the results must be carefully interpreted.
[Reply]
As you have pointed out, data that was collected at a single point of time was used for analysis. This is one of the most critical limitations in this study. Any parameter would have fluctuated throughout the observation period. Particularly, for patients with long survival periods, it was impossible to predict the tendency of the parameters, including serum phosphate levels, using the single-point data. We have emphasized this limitation in the Discussion section.
Moreover, as another reviewer suggested, we considered the possibility of following up on the serum phosphate levels. However, as the number of patients decreased after the inclusion period on account of their deaths, we conducted the linear regression model analysis between serum phosphate levels during the inclusion period and those recorded one year after the inclusion period (n=258). Although the serum phosphate levels one year after the inclusion period correlated with those at the inclusion, the correlation was not so strong (r=0.47, p<0.001), suggesting that serum phosphate levels had fluctuated during the following period. This correlational analysis has been included in the methods, results, and discussion sections. We appreciate your opinion.
In conclusion, it should rather be written: serum phosphate levels may play an important role in the development 234 of AS in HD patients.
[Reply]
We appreciate your suggestion. We have accordingly revised the conclusion.
Also, it is know that the concentration of phosphate in hemodialysis patients depends on the dialysis dose. There is no information on dialysis in the article, what was the treatment regimen? What is the adequacy of dialysis (Kt / V?). You need to add this information.
[Reply]
Thank you for your reasonable comment. We have added the data for Kt/V in Table 1. However, there was no significant difference in Kt/V between patients that developed AS and patients that did not develop AS.

Reviewer 2 Report
The study by Torigoe et al investigates the prognostic role of serum phosphate levels on the development of severe aortic stenosis (AS) in patients with renal replacement therapy including 302 patients. Besides age, especially serum phosphate levels increased the risk of developing severe AS. The manuscript is clinically relevant and well-written. The following comments are made by this reviewer.
Specific comments:
- Results: What is meant with “… or were lost to follow-up before their birthdays”. Please clarify. I suggest to avoid the term “birthdays”, it may be confusing for some readers.
- Can the authors provide phosphate levels during follow up? This would prove the robustness of the data. Although the authors mention they have only a single phosphate sample, they should try to provide follow-up phosphate levels.
- Propensity score matching should be done for the comparison of AS versus no AS, there are significant differences regarding the baseline characteristics. This would reduce the chance of possible selection bias.
Author Response
Reviewer 2
The study by Torigoe et al investigates the prognostic role of serum phosphate levels on the development of severe aortic stenosis (AS) in patients with renal replacement therapy including 302 patients. Besides age, especially serum phosphate levels increased the risk of developing severe AS. The manuscript is clinically relevant and well-written. The following comments are made by this reviewer.
Specific comments:
Results: What is meant with “… or were lost to follow-up before their birthdays”. Please clarify. I suggest to avoid the term “birthdays”, it may be confusing for some readers.
[Reply]
We are sorry that the expression was difficult to understand. As the data was collected in patients’ birth months, we wanted to convey that patients who did not undergo hemodialysis in their birth months during the study period from 2011 July to 2012 June were excluded from this study. Therefore, we have revised this line in the manuscript as follows on lines 90–91:
“Of 441 patients undergoing HD during the inclusion period, patients whose annual routine examination data could not be collected during this period were excluded”.
Can the authors provide phosphate levels during follow up? This would prove the robustness of the data. Although the authors mention they have only a single phosphate sample, they should try to provide follow-up phosphate levels.
[Reply]
Thank you for this suggestion to provide follow-up phosphate levels. However, in the case of our study, recording follow-up serum phosphate levels was difficult because of the already large requirement for data collection during the observation period. In addition, some patients died soon after the inclusion and the availability of data varied depending on the patients’ survival period. Therefore, we have only used data collected at a single-point in this study.
For example, 44 out of 302 patients died within one year, and 258 patients could be followed up one year after inclusion. The mean of serum phosphate levels one year after the inclusion period was 5.4±1.7 mg/dL. The linear regression analysis showed that there was a moderate correlation between serum phosphate levels during the inclusion period with those recorded one year after the inclusion (r=0.47, p<0.001). We have added this fact in the result section and have revised the methods. Using only a single-point of serum phosphate level data is one of the most critical limitations in this study, and the correlation was not so strong. Therefore, we have included this as a limitation in the discussion section. Thank you.
Propensity score matching should be done for the comparison of AS versus no AS, there are significant differences regarding the baseline characteristics. This would reduce the chance of possible selection bias.
[Reply]
Thank you for suggestion to perform propensity score matching. Although we conducted a survival analysis, the main outcome of this study is the development of AS. Therefore, we are afraid that conducting propensity score matching for patients that developed AS vs those that did not develop AS would not significantly support our argument. In addition, we have conducted this study in a complete enumeration manner, which means we included eligible patients as many as possible in our hospital. Therefore, in our humble opinion, we do not think that there is a selection bias of the included patients in this study.
